# Predictors of metabolic syndrome among teachers in under-resourced schools in South Africa: Baseline findings from the *KaziHealth* workplace health intervention

Nandi Joubert[1,2,3,4]*, Larissa Adams[4], Jan Hattendorf[1,2], Jan Degen[3], Danielle Dolley[4], Annelie Gresse[5], Ivan Müller[3], Siphesihle Nqweniso[4], Nicole Probst-Hensch[1,2], Harald Seelig[3], Peter Steinmann[1,2], Rosa du Randt[4], Uwe Pühse[3], Jürg Utzinger[1,2‡], Markus Gerber[3‡], Cheryl Walter[4‡]

1. Swiss Tropical and Public Health Institute, Allschwil, Switzerland, 2. University of Basel, Basel, Switzerland, 3. Department of Sport, Exercise and Health, University of Basel, Basel, Switzerland, 4. Department of Human Movement Science, Nelson Mandela University, Gqeberha, South Africa, 5. Department of Human Nutrition and Dietetics, Nelson Mandela University, Gqeberha, South Africa

‡ Authors share senior authorship
* nandi.joubert@swisstph.ch

## Abstract

Non-communicable diseases (NCDs) account for most of global deaths, with rising prevalence in low-and middle-income countries. Metabolic syndrome (MetS) is a cluster of interrelated NCD risk factors, including obesity, dyslipidaemia, hypertension, and hyperglycaemia, that amplify NCD risk. Four leading modifiable factors driving NCDs are physical inactivity, tobacco and alcohol use, and an unhealthy diet. Teachers in under-resourced schools face significant health challenges, with their wellbeing often overlooked in South Africa. The prevalence and severity of MetS, adherence to physical activity guidelines, and associations with modifiable NCD risk factors were assessed. This study, part of the '*KaziBantu*: Healthy Schools for Healthy Communities' project, included 168 teachers (aged 21–72, mean = 47 years) from 8 under-resourced schools in Gqeberha, South Africa. Data collection included MetS markers, device-measured physical activity, tobacco and alcohol use, and total fat, saturated fat, sugar, and sodium intake. Covariates included age, sex, race, education, and household income. MetS was observed in 58% of participants, with central obesity (79%) being the most prevalent component, followed by hypertension (59%). The largest proportion of participants (26%) had 3 MetS components, followed by 4 components (20%), while 12% had all 5 components. Nearly half (44%) of teachers were physically inactive and 53% exceeded total fat intake recommendations. Bayesian multilevel logistic regression revealed key predictors of MetS: age (per year increase; odds ratio[OR]=1.15, 95% credible interval[CrI] [1.07, 1.24]) and high daily total fat intake (OR=3.37, 95%CrI [1.03, 11.96]) were positively associated, while hours spent in moderate-to-vigorous intensity physical activity per week (OR=0.73, 95%CrI [0.55,

**Data availability statement:** The dataset underlying this study is publicly available on Zenodo at (https://doi.org/10.5281/zenodo.15072311). The dataset includes all variables necessary to replicate the study's findings and is shared under a Creative Commons Attribution 4.0 International (CC-BY 4.0) license.

**Funding:** This research was financially supported by the Novartis Foundation (Basel, Switzerland; grant no. 1001 011 45) and the South African Medical Research Council (SAMREC) (Cape Town, South Africa; grant no. 616268). The funders had no role in the study design, data collection, data analysis or data interpretation, nor the decision to submit the paper for publication. This research took place under the auspices of the UNESCO Chair on Physical Activity and Health in Educational Settings (https://unesco-chair.dsbg.unibas.ch/en/).

**Competing interests:** The authors have declared that no competing interests exist.

0.95]) and higher monthly household income (OR=0.14, 95%CrI [0.02, 0.72]) were protective. Addressing physical inactivity and unhealthy diets through tailored interventions is crucial to reducing MetS prevalence and improving the health of teachers working in low-resourced settings. Effective solutions should empower healthier lifestyles while tackling structural barriers to health equity.

## 1. Introduction

Non-communicable diseases (NCDs) are chronic, non-infectious diseases caused by a complex interplay of genetic, physiologic, lifestyle, socioeconomic, and environmental factors [1]. In 2021, NCDs caused 43 million deaths (75% of all deaths globally), and were the primary cause of disability and premature death (death before the age of 70 years) worldwide [2]. The share of the total disease burden due to NCDs has increased globally from 43.2% in 1990 to 63.8% in 2019 [3]. This increase has not decelerated and, without major intervention, NCDs are estimated to increase further by 27% in sub-Saharan Africa alone over the next decade [2].

The burden of NCDs is particularly high in low- and middle-income countries (LMICs), accounting for 73% of all NCD deaths and 82% of premature deaths [2]. At current rates, NCDs are further projected to surpass infectious, maternal and child, and undernutrition-related diseases to become the leading cause of disability in Africa by 2030 [4]. South Africa, though categorized as an upper-middle-income country, carries a high burden of NCDs, while also facing unique challenges arising from its history of apartheid and persisting inequality. The country ranks among those with the highest and most persistent inequality levels globally, with race remaining a central driver of inequality.

The World Health Organization (WHO) has identified four major lifestyle-related risk factors for the development of NCDs: (i) physical inactivity; (ii) tobacco consumption; (iii) harmful alcohol use; and (iv) an unhealthy diet [2]. Since these risk factors are modifiable through behavioural interventions, it is estimated that a large proportion of NCDs could be prevented by reducing these [2]. In many LMICs, however, the majority of healthcare resources are required for treatment, leaving little for prevention and education strategies, even though these could prevent and lower the burden of disease, and may be more cost-effective in the long run [5]. Physical inactivity has been identified as the fourth leading risk factor for mortality and a major contributor to the development of NCDs [6]. Physical inactivity is estimated to be more than twice as high in high-income countries (HICs) compared to LMICs (36.8% *vs.* 16.2%) [7]. However, measuring physical activity in LMICs is challenging, mainly due to resource constraints, resulting in subjective self-report techniques (such as questionnaires and diary logs) being employed to estimate activity, which not only hinders comparability with device-based activity measurements done in HICs but also complicates understanding activity patterns and the nature of activity in LMICs. Technological advancements in device-based physical activity measurement have resulted in open-source and raw data processing techniques becoming more accessible and popular compared to black-box

manufacturer analysis methods, as more flexibility, accuracy, and advanced analytics result in more comparable outputs across studies. As open-source accelerometry is quite complex and time- and resource-intensive, it has not been the standard of analysis when device-based activity measurements have been employed in LMICs. Hence, open-source accelerometry research in LMICs may provide new insights and highlight specific intervention needs.

Metabolic syndrome (MetS), a cluster of NCD risk factors that include central obesity, dyslipidaemia, hypertension, and hyperglycaemia, occur together more often than alone, are interrelated, and share underlying mediators, mechanisms, and pathways [8]. MetS increases the risk of NCDs, especially cardiovascular disease, which account for most of NCD deaths [2]. Regardless of MetS diagnostic criteria, HICs show the highest prevalence of MetS [9]. That said, although sub-Saharan Africa comprises 13.5% of the global population, it contributes less than 1% of the world's research data [10]. This holds true for research pertaining to MetS: a global meta-analysis investigating the geographic distribution of MetS showed that studies conducted in Africa contributed the least at 7% (78 studies) of a total of 1,129 studies worldwide [11]. It follows that more research is required in the African region to comprehensively understand the prevalence, burden, and development of MetS, shedding new light on a major contributor to NCDs.

Teachers play a pivotal role in society, significantly influencing both the academic and personal development of future generations. In community settings that face poverty and related social ills, the role of a teacher becomes even more important, as research consistently shows that children who succeed despite many challenges typically have at least one stable, supportive relationship with an adult – often a teacher [12]. However, teachers in under-resourced public schools in South Africa face unique challenges, including overcrowded classrooms (teacher-learner ratio: > 1:39), dilapidated infrastructure, safety concerns, and limited access to adequate sanitation, teacher support, and materials [13]. These challenges lead to physical and psychological stressors among educators. Although health and lifestyle-related information for public teachers in South Africa is limited, work-related factors have been shown to result in high levels of stress and stress-related illnesses in a large-scale, nationally representative sample of public school teachers [14]. Assessing and addressing the health of teachers in these settings is essential not only for their individual wellbeing but also for broader public health and educational outcomes, as healthy teachers are more effective in their roles and can cultivate resilient, engaged learners while promoting wellbeing and empowering community environments.

The purpose of this study was three-fold. First, to assess the prevalence and severity of MetS in an under-researched sample of public primary school teachers, who teach in low-resource settings in South Africa. Second, to determine whether the latest WHO physical activity guidelines were met using device-based measurement with open-source acceleration data processing. Third, to investigate the association between MetS and the leading modifiable risk factors for NCDs.

## 2. Materials and methods

### 2.1. Ethical statement

The *KaziBantu* project was conducted in accordance with ethical principles outlined in the latest Declaration of Helsinki. The study received ethical approval from the following committees before the study commenced: (i) Nelson Mandela University Research Ethics Committee (Human) on 26 March 2018 (H19-HEA-HMS-001); (ii) Eastern Cape Department of Health on 5 June 2018 (EC_201804_007); (iii) Eastern Cape Department of Education on 9 May 2018 (no reference number provided); and (iv) Ethics Committee Northwest and Central Switzerland (EKNZ) on 1 March 2018 (R-2018–00047). The study was additionally registered with the International Standard Randomised Controlled Trial Number (ISRCTN) Registry (ISRCTN18485542). Informed written consent was obtained from all participants before data collection, after clear explanations of the study's purpose, risks, and benefits. Participant data were anonymised and handled securely to maintain privacy and confidentiality throughout and after the study. All procedures were conducted by qualified personnel trained in research ethics.

## 2.2. Study design

This research is embedded in '*KaziBantu*: Healthy Schools for Healthy Communities', a cluster-randomized controlled trial that investigated the effects of a school-based holistic health intervention program in under-resourced primary schools in Gqeberha, in the Eastern Cape province of South Africa, between 2019 and 2022. The protocol has been published elsewhere [15]. *KaziBantu* had two intervention arms: *KaziKidz* focused on promoting health and activity in children, while *KaziHealth*, an embedded exploratory design, focused on teachers' physical and mental health through a workplace program. The *KaziHealth* baseline assessments were the source data for this manuscript.

## 2.3. Setting and participants

The *KaziBantu* project was conducted within under-resourced public primary schools situated in peri-urban areas in Gqeberha (formerly Port Elizabeth), in the Nelson Mandela Bay Metropole, in the Eastern Cape province of South Africa. The Eastern Cape province remains the poorest province and has the highest unemployment and crime rates in the country. In 2024, the Nelson Mandela Bay Metropole was ranked as the most dangerous city in South Africa and ranked 9th globally [16].

Eight quintile 3 primary schools formed part of the study. The South African school system categorises public schools into 5 quintiles for the allocation of financial resources based on national poverty tables, with quintile 1 being the poorest, and quintile 5 being the least poor [17]. Quintile 3 primary schools are the lowest quintile found within the Nelson Mandela Bay area. Four schools were located in township areas and 4 schools in the Northern areas of Gqeberha. Townships in South Africa refer to underdeveloped urban informal settlements that were historically reserved for non-White populations during apartheid and are typically located on the periphery of cities. In the Nelson Mandela Bay area, the 'Northern areas' were designated as residential zones for the Coloured population, a distinct racial category in use within official and social contexts in South Africa. Although rooted in the historical and political framework of South Africa's colonial and apartheid past, the term represents a complex and multifaceted identity that continues to be embraced today, reflecting a rich, multifaceted heritage and a deep sense of belonging. Both townships and the Northern areas are characterised by limited access to basic services and infrastructure, as well as significant socioeconomic challenges. Notably, the Nelson Mandela Bay Metropolitan remains the most segregated city among South Africa's largest metropolitan areas [18].

School selection started with principals from quintile 3 primary schools being invited to attend an information meeting held at the Eastern Cape Department of Education in October 2018. Principals from 349 schools were invited to the meeting, of which 64 schools requested further project information. Once further project details had been shared, and schools had been visited, 8 schools were selected based on being a typical quintile 3 school, geographic location (peri-urban area), spoken language (English, Afrikaans, or Xhosa), and commitment by school principals to support the project activities. All teachers in the 8 schools were invited to participate in the *KaziHealth* workplace health intervention program, with participant recruitment conducted between 16 January and 30 April 2019.

## 2.4. Assessments

**2.4.1. Demographic information.** Self-reported demographic indicators included date of birth (to establish age in years in 2019), sex, and race. We included collecting race information as most of our study sample belong to groups that have been systematically oppressed and disadvantaged under South Africa's apartheid and colonial legacies and such information can uncover health inequities. Race is a social construct and not a fixed biological trait, and hence, race differences are seen in light of the wider socioeconomic context and structural disparities. Race was reported based on official categories in the South African context, and included: Black, Coloured and Indian, White, 'other', and 'prefer not to say'. Participants were further asked to report on educational attainment (diploma, university degree, or postgraduate

degree) and socioeconomic status (SES). In this study, SES was operationalised using average monthly household income, categorised as lower (≤R 20,000), middle (R 20,001-40,000), or higher (>R 40,000), in line with teachers' salary scales in South Africa. This should be interpreted as a proxy for SES rather than a multidimensional construct. Teachers were also asked to report medication taken for dyslipidaemia, hypertension, or hyperglycaemia, as this information was necessary to accurately determine MetS status.

**2.4.2. Metabolic syndrome.** Several MetS diagnostic criteria have been proposed over the past decades; by WHO in 1998, the European Group for the Study of Insulin Resistance in 1999, the Adult Treatment Panel III of the National Cholesterol Education Program in 2001, the American Heart Association/National Heart, Lung, and Blood Institute (AHA/NHLBI) in 2004, the International Diabetes Federation (IDF) in 2005, and a harmonising agreement between the AHA/NHLBI and IDF, known as the Joint Interim Statement (JIS) in 2009 [9,19–23]. The latest MetS definition, which was used in this study to establish MetS status, diagnoses MetS when 3 out of the following 5 criteria are met: (i) population-specific waist circumference: ≥94 cm for men and ≥80 cm for women in sub-Saharan African, European, Middle East, and Mediterranean populations and ≥90 cm for men and ≥80 cm for women in Asian, Japanese, and Ethnic South and Central American populations; (ii) fasting triglyceride levels: ≥1.7 mmol/L or on treatment for dyslipidaemia; (iii) high-density lipoprotein cholesterol (HDL-C): <1.0 mmol/L in men and <1.3 mmol/L in women or on treatment for dyslipidaemia; (iv) systolic blood pressure (SBP): ≥130 mmHg and/or diastolic blood pressure (DBP): ≥85 mmHg or on treatment for hypertension; and (v) fasting glucose: ≥5.6 mmol/L (equivalent to glycated haemoglobin [HbA1c] ≥5.7%, which has shown comparable performance to fasting plasma glucose in the South African context) or on treatment for hyperglycaemia [9,24].

Waist circumferences were measured to determine abdominal obesity. Measurement were to the nearest 0.1 cm, using a flexible, non-extensible steel tape (Seca 201, Surgical SA; Johannesburg, South Africa), using standard guidelines and classifications [25]. Further, body composition analysis included height, weight, and body mass index (BMI), using standard guidelines and classifications [25]. Body weight, to the nearest 0.1 kg, was measured with a bioelectrical impedance scale (Tanita MC-580, Tanita Corp.; Tokyo, Japan). Body height was measured with a Seca stadiometer (Surgical SA; Johannesburg, South Africa), to the nearest 0.1 cm.

Blood pressure was assessed with a validated oscillometric digital blood pressure monitor (Omron M6 AC; Hoofddorp, Netherlands). Participants were seated quietly for 5 min before 3 blood pressure measurements were taken from the left arm at heart level, with a 1-min interval between measurements, using an appropriate cuff size, based on arm circumference. As indication of SBP and DBP, the mean (M) of the last two measurements were used with standard classification [26].

Overnight fasting capillary sampling, using established techniques and classifications, were used to assess blood lipids (including triglycerides and HDL-C) and HbA1c with the point-of-care Alere Afinion AS100 analyser (Abbott Technologies; Abbott Park; Illinois, USA) [27–29]. The Alere Afinion AS100 analyser's accuracy and utility in community settings has been described elsewhere [30].

A referral for further medical screening was provided for any elevated metabolic markers identified. Public school teachers in South Africa have access to health insurance through the employer subsidy to the Government Employees Medical Scheme (GEMS), facilitating follow-up healthcare and further health assessments.

**2.4.3. Leading modifiable risk factors for NCDs.** Physical inactivity: Device-based physical activity was measured through open-source accelerometry with research grade tri-axial ActiGraph wGT3X-BT accelerometers (firmware 1.9.2) (ActiGraph; Pensacola, Florida, USA). Accelerometers were initialised with ActiLife software (version 6.13.4) with a sample rate of 30 Hz, idle sleep mode enabled, and local time used (GMT+2). Participants were instructed to wear the accelerometer around the hip for 7 consecutive days (5 weekdays and 2 weekend days), only taking the accelerometer off during sleep and when encountering water. The raw acceleration data were processed with GGIR (version 3.1-4), an open-source R package [31,32]. GGIR converts acceleration magnitude into Euclidian norm minus 1, with negative

values set to zero (ENMONZ) by subtracting 1 gravitational equivalent unit ($g$) [1 $g$ = 9.81 m/s²] from vector magnitude to remove the gravity component of the acceleration signal and isolate the activity-related signal, expressed in milligravity (m$g$) and calculated with the following equation: $ENMONZ\ (mg) = \sqrt{acc_x^2 + acc_y^2 + acc_z^2} - 1$ where x, y, and z axes are accelerometer measured planes of movement.

Files were analysed with 60 sec epochs with time from midnight to midnight (24-h). Days with wear time ≥8 h were considered valid. Zero counts for ≥60 min was defined as non-wear time. Mean acceleration (ENMONZ) is presented in m$g$ in min/day and time (min/week) spent in moderate-to-vigorous intensity physical activity (MVPA), identified by mean acceleration above threshold of 70 m$g$, for bout duration of 1 min, and inclusion criterion of more than 80% [33,34]. The latter was used to determine whether the WHO physical activity guidelines of a minimum of 150 min of MVPA per week were met [35].

**Substance use:** Data on substance use were self-reported and included: (i) current tobacco use or cessation within the past 6 months and (ii) alcohol consumption. As no safe amount of alcohol consumption for health can be established, no quantity or frequency was specified for alcohol intake [36].

**Unhealthy diet:** Dietary analysis was conducted using a single weekday 24-h dietary recall (24HR) structured interview, to obtain food and beverage consumption in a 24-h period, including portion sizes and preparation methods [37]. 24HR were analysed to obtain energy and nutrient intake with the Medical Research Council of South Africa (MRCSA) Food-Finder version 3.0 (MRCSA; Cape Town, South Africa) [38]. Relevant nutrient intake per day was assessed based on the WHO healthy diet guidelines, and included: (i) total fat intake <30% of total daily energy intake; (ii) saturated fat intake <10% of total daily energy intake; (iii) sodium intake ≤2,000 mg; and (iv) sugar intake <10% of total daily energy intake and <50 g daily intake [39].

## 2.5. Statistical analysis

For descriptive statistics, sample size (n), median (Med), and interquartile range (IQR) are reported for continuous variables, while frequencies (n) and percentages (%) are reported for categorical variables. The association between MetS, demographic factors and modifiable risk factors for NCDs, was assessed using a multilevel Bayesian logistic regression model. All analyses were performed with R version 4.2.3 (R Core Team; Vienna, Austria).

### 2.5.1. Missing data imputation.
Waist circumference data were missing for 19% of the observations. Given the importance of this variable in establishing MetS status, we employed multiple imputation by chained equations (MICE) using the 'mice' package in R (version 3.16.0), with predictive mean matching (PMM) [40]. The imputation model included height, weight, BMI, age, and sex, and was performed with 20 imputations and 200 iterations per imputation. Convergence diagnostics, particularly the Gelman-Rubin statistic ($\hat{R}$), was used to assess whether the imputation had stabilized. To assess the imputation, we compared descriptive statistics and correlation matrices of the original and imputed datasets to ensure that the imputation did not introduce any systematic bias and that the relationships between variables were preserved across the original and imputed datasets. We generated only 1 imputed dataset for the subsequent analyses, and we acknowledge that this approach does not carry forward the uncertainty from imputation into the Bayesian model. The resulting dataset, with imputed values, was used in subsequent analyses and reporting of results.

### 2.5.2. Multilevel Bayesian logistic regression.
A multilevel Bayesian logistic regression model was used to assess the association between the outcome, MetS, demographic factors (age, sex, race, educational attainment, and SES through household income) and the WHO leading modifiable risk factors for NCDs. A Bayesian model was chosen

as it facilitates more robust estimation in the presence of sparse data. The model was fitted using the R package 'brms' (version 2.22.0), which interfaces with Stan (StanHeaders version 2.32.10, rstan version 2.32.6) [41,42].

We employed a multilevel model accounting for the natural grouping of participants within the 8 schools. We further implemented regularization via weakly informative priors, which helps mitigate overfitting, especially in the presence of multiple predictors with limited observations. A normal prior (M = 0, standard deviation (SD) = 10) was assigned to the intercept, while a Cauchy prior (location = 0, scale = 2.5) was applied to fixed effects.

The multilevel Bayesian logistic model was fitted with 4 chains of 10,000 iterations each (including 2,000 warm-up iterations), and the posterior was sampled using the No-U-Turn Sampler (NUTS). We set a maximum tree depth of 15 and an adapt delta of 0.999 to ensure stable sampling and minimize divergence issues. The model used a binomial likelihood for logistic regression, specified through the Bernoulli family, which is appropriate for the binary outcome. A logit link function was applied to model the log-odds of the binary outcome as a linear combination of predictors. Model diagnostics demonstrated excellent convergence and sampling reliability, confirming its suitability. Gelman-Rubin statistic ($\hat{R}$) for all parameters were between 0.99 and 1.01, indicating strong convergence across chains. Effective sample sizes (ESS) were uniformly high, with bulk ESS values exceeding 1,000, ensuring robust and reliable estimates for central tendencies, while tail ESS supported accurate uncertainty estimation. Trace plots showed well-mixed chains without drift or divergence, reinforcing the model's stability. Posterior predictive checks (PPC) revealed strong alignment between observed and posterior predicted distributions, suggesting that the model captured the data's underlying structure effectively. The pre-specified model was designed to achieve an optimal balance between predictive accuracy and parsimony, making it well-suited for inference.

## 3. Results

### 3.1. Characteristics of participants

Teachers forming part of the study represented 82% of the eligible teachers from the selected schools, totalling 168 participants. As normally seen in primary school teaching staff in South Africa, the majority (85%, n = 142) were women, with men representing 15% (n = 26). Age ranged from 21 to 72 years (M = 47.2, SD = 10.7 years). Descriptive statistics for all study variables are presented in Tables 1 (continuous variables andcategorical variables) for the entire study sample.

### 3.2. Metabolic syndrome prevalence and severity

Of the 158 participants with complete datasets, 58% (n = 92) were classified with MetS, meeting the latest criteria from the JIS [9]. When considering the individual components constituting MetS, central obesity was the most prevalent with 79% (n = 132), followed by hypertension 59% (n = 92), high triglycerides 51% (n = 82), low HDL-C 43% (n = 69), and hyperglycaemia 40% (n = 66). These results include participants taking medication for hypertension, dyslipidaemia, and hyperglycaemia.

Fig 1 displays the distribution of MetS components by severity, showing the proportion of participants with 0–5 MetS components. The results indicate that the minority of participants (8%, n = 12) had none of the components, whereas the largest proportion (26%, n = 38) had 3 components. Notably, 20% (n = 30) exhibited 4 components, and 12% (n = 18) had all 5 MetS components, underscoring the high level of metabolic dysfunction in the study sample. The stacked bars for each component count are ordered by prevalence, with central obesity, hypertension, and high triglycerides consistently appearing as the most common MetS components across component counts, signifying their contribution to the overall MetS burden in this population.

The detailed distribution of MetS components and their various combinations among participants is presented in the supplementary material, S1 Fig. This figure provides a comprehensive view of the prevalence and overlap of individual components, highlighting the MetS clustering patterns in the study population.

**Table 1. Descriptive statistics for all study variables among public primary school teachers in mid-2019 in Gqeberha, South Africa.**

| Continuous variables | Sample size (n)[1] | Median (Med) | Interquartile range (IQR) |
|---|---|---|---|
| **Demographics** | | | |
| Age (years in 2019) | 168 | 50.0 | 43.8–54.3 |
| **Metabolic biomarkers** | | | |
| Metabolic syndrome component count[2] | 149 | 3 | 2 - 4 |
| Triglycerides (mmol/L) | 162 | 1.60 | 1.14–2.20 |
| HDL-C[3] (mmol/L) | 160 | 1.37 | 1.16–1.62 |
| Systolic blood pressure (mmHg) | 157 | 126 | 114–140 |
| Diastolic blood pressure (mmHg) | 157 | 84 | 77–92 |
| HbA1c[4] (%) | 166 | 5.5 | 5.3–5.9 |
| **Body composition** | | | |
| Height (cm) | 165 | 161.4 | 155.8–165.3 |
| Weight (kg) | 165 | 84.6 | 72.0–96.2 |
| Body mass index (kg/m²) | 165 | 32.6 | 27.2–38.3 |
| Waist circumference[5] (cm) | 168 | 97.8 | 85.1–106.0 |
| **Physical activity** | | | |
| ENMONZ[6] (m$g$[7] in min/day) | 144 | 10.57 | 9.16–12.08 |
| MVPA[8] (m$g$[7] in min/week) | 144 | 160.12 | 97.02–227.76 |
| **Nutrient intake (per day)** | | | |
| Total fat (g) | 139 | 65 | 45–90 |
| Saturated fat (g) | 139 | 19 | 14–27 |
| Total sugar (g) | 139 | 60 | 39–81 |
| Sodium (mg) | 139 | 1495 | 1015–2354 |
| **Categorical variables** | Category | Frequency (n)[1] | Percentage (%)[9] |
| **Demographics** | | | |
| Area (school location) | Northern area | 168 | |
| | | 80 | 47.6 |
| | Township area | 88 | 52.4 |
| School | | 168 | |
| | 1 | 18 | 10.7 |
| | 2 | 30 | 17.9 |
| | 3 | 22 | 13.1 |
| | 4 | 17 | 10.1 |
| | 5 | 15 | 8.9 |
| | 6 | 18 | 10.7 |
| | 7 | 27 | 16.1 |
| | 8 | 21 | 12.5 |
| Sex | Man | 168 | |
| | | 26 | 15.5 |
| | Woman | 142 | 84.5 |
| Race | Black | 168 | |
| | | 89 | 53.0 |
| | Coloured and Indian | 72 | 42.9 |
| | White | 7 | 4.2 |
| Tertiary education attainment | Diploma | 162 | |
| | | 83 | 51.2 |

*(Continued)*

| Continuous variables | Sample size (n)[1] | Median (Med) | Interquartile range (IQR) |
|---|---|---|---|
| | University degree | 54 | 33.3 |
| | Postgraduate degree | 25 | 15.4 |
| Socioeconomic status based on average monthly household income | Lower: ≤ R 20 000 | 168 | |
| | | 70 | 41.7 |
| | Middle: R 20 000–R 40 000 | 71 | 42.3 |
| | Higher: > R 40 000 | 27 | 16.1 |
| Medication use for dyslipidaemia | | 168 | |
| | No | 153 | 91.1 |
| | Yes | 15 | 8.9 |
| Medication use for hypertension | | 168 | |
| | No | 147 | 87.5 |
| | Yes | 21 | 12.5 |
| Medication use for hyperglycaemia | | 168 | 90.5 |
| | No | 152 | |
| | Yes | 16 | 9.5 |
| **Metabolic biomarkers** | | | |
| Metabolic syndrome [10] | | 158 | 41.8 |
| | No | 66 | |
| | Yes | 92 | 58.2 |
| Metabolic syndrome criteria for waist circumference:[5] | | 168 | 21.4 |
| ≥ 94 cm for men, ≥ 80 cm for women in sub-Saharan African | No | 36 | |
| | Yes | 132 | 78.6 |
| Metabolic syndrome criteria for triglycerides: | | 162 | |
| ≥ 1.7 mmol/L or on treatment for dyslipidaemia | No | 80 | 49.4 |
| | Yes | 82 | 50.6 |
| Metabolic syndrome criteria for HDL-C[3]: <1.0 mmol/L for men, <1.3 mmol/L for women or on treatment for dyslipidaemia | | 160 | |
| | No | 91 | 56.9 |
| | Yes | 69 | 43.1 |
| Metabolic syndrome criteria for blood pressure: systolic ≥130 mmHg and/or diastolic ≥85 mmHg or on treatment for hypertension | | 157 | |
| | No | 65 | 41.4 |
| | Yes | 92 | 58.6 |
| Metabolic syndrome criteria for glucose: fasting glucose ≥5.6 mmol/L (equivalent to HbA1c[4] ≥ 5.7%) or on treatment for hyperglycaemia | | 167 | |
| | No | 101 | 60.5 |
| | Yes | 66 | 39.5 |
| **Substance use** | | | |
| Tobacco use: "Do you currently smoke tobacco products, or have you quit smoking in the past 6 months?" | | 168 | |
| | No | 142 | 84.5 |
| | Yes | 26 | 15.5 |
| Alcohol use: "Do you consume alcohol?" | | 167 | |
| | No | 108 | 64.7 |
| | Yes | 59 | 35.3 |

*(Continued)*

**Table 1.** (Continued)

| Continuous variables | Sample size (n)[1] | | Median (Med) | Interquartile range (IQR) |
|---|---|---|---|---|
| **Physical activity** | | | 144 | |
| Met WHO physical activity guideline[11] | No | | 63 | 43.8 |
| | Yes | | 81 | 56.3 |
| **Nutrient intake (per day)[12]** | | | | |
| High total fat | | | 139 | |
| | No | | 66 | 47.5 |
| | Yes | | 73 | 52.5 |
| High saturated fat | | | 139 | |
| | No | | 80 | 57.6 |
| | Yes | | 59 | 42.4 |
| High total sugar | | | 139 | |
| | No | | 50 | 36.0 |
| | Yes | | 89 | 64.0 |
| High sodium | | | 139 | |
| | No | | 97 | 69.8 |
| | Yes | | 42 | 30.2 |

**Legend for Table 1:**

[1]Number of participants varies due to different numbers of missing values in specific variables

[2]Number of positive metabolic syndrome components: central obesity[5], high triglycerides, low HDL-C[3], hypertension, and/or hyperglycaemia

[3]HDL-C: High-density lipoprotein cholesterol

[4]HbA1c: Glycosylated haemoglobin

[5]Imputed waist circumference values included

[6]ENMONZ: Mean acceleration: Euclidian norm minus one, with negative values set to zero

[7]m$g$: Milligravity

[8]MVPA: Moderate-to-vigorous intensity physical activity, mean acceleration above threshold of 70 m$g$[7], for bout duration of 1 min, and inclusion criterion of 80%

[9]Percentage (%) calculated based on non-missing data for each variable

[10]Metabolic syndrome if 3/5 criteria is present (central obesity[5], high triglycerides, low HDL-C[3], hypertension, and/or hyperglycaemia), defined by the Joint Interim Statement

[11]Met World Health Organization (WHO) physical activity guideline: minimum 150 min of MVPA[8] per week

[12]World Health Organization (WHO) diet guidelines: high total fat intake ≥30%; high saturated fat intake ≥10%; high sodium intake >2000 mg; and high sugar intake ≥10% and/or ≥50 g of total daily energy intake

### 3.3. Leading modifiable risk factors for NCDs

**3.3.1. Physical inactivity.** Accelerometer data were available for 144 participants, 85% of the total sample. A total of 23 participants did not want to wear the accelerometer device, and 1 participant failed to meet the wear time criteria of ≥8 h per day. Of the participants with valid accelerometer data, 56% (n = 81) met the latest WHO physical activity guidelines with at least 150 min of MVPA per week, whereas the remaining 44% (n = 63) did not meet this threshold.

**3.3.2. Substance use.** Information on tobacco use was available for all participants (n = 168), with 15% (n = 26) indicating that they currently smoke tobacco products or have quit within the past 6 months (n = 4). Information on alcohol use was available for 167 participants and was reported in 35% (n = 59) of the participants.

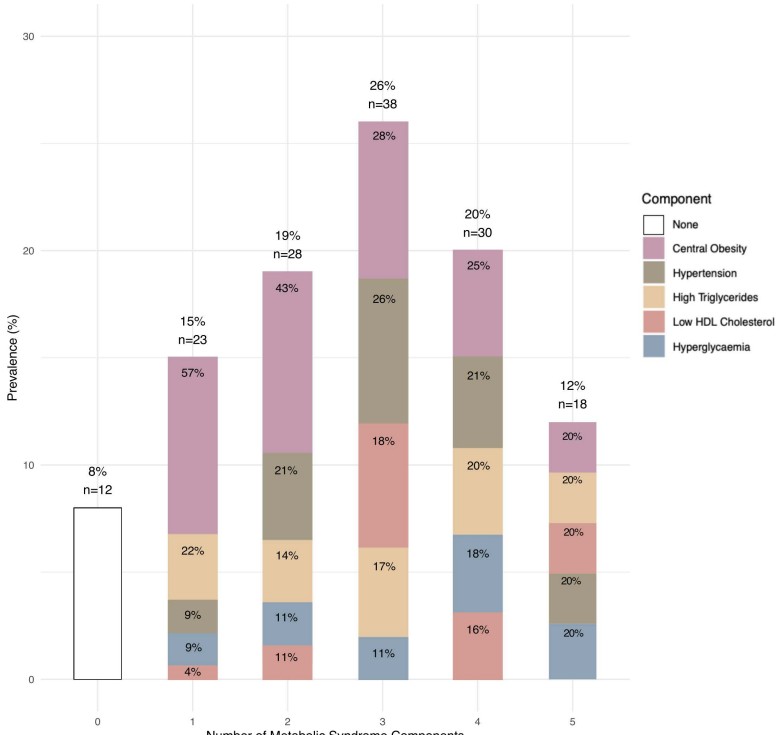

**Fig 1. Distribution of MetS components by severity count among 149 public primary school teachers in mid-2019 in Gqeberha, South Africa.**

**3.3.3. Unhealthy diet.** Key nutrient intake constituting an unhealthy diet, according to WHO, include high total fat, saturated fat, sodium, and sugar intake [39]. Dietary information was available for 83% (n = 139) of the participants. Of these, more than half (53%) consumed a high total fat diet, while less than half (42%) consumed a high saturated fat diet. High sugar intake was reported by 64% of participants, and 30% consumed a diet high in sodium.

### 3.4. MetS and leading modifiable risk factors for NCDs

Table 2 presents the results of the multilevel Bayesian logistic regression model, which identified significant predictors of MetS.

Age was positively associated with MetS, with each additional year increasing the odds by 15% (OR = 1.15, 95% CrI [1.07, 1.24]). High daily total fat intake was associated with over a threefold increase in the odds of MetS (OR = 3.37, 95% CrI [1.03, 11.96]), highlighting it as a significant risk factor. Physical activity was found to be protective, with each additional hour of MVPA per week reducing the odds of MetS by 27% (OR = 0.73, 95% CrI [0.55, 0.95]). Higher SES (monthly household income >R 40 000) was further found to be protective, with significantly lower odds of MetS compared to incomes ≤R 20 000 (OR = 0.14, 95% CrI [0.02, 0.72]).

Fig 2 provides a visual summary of the associations between MetS and its predictors, as identified by the multilevel Bayesian logistic regression model (Table 2). Fig 2 highlights the relative strength and direction of associations for demographic, behavioural, and dietary factors. The ORs, presented on a logarithmic scale alongside credible intervals (95% CrI), emphasise the variability in effect sizes and the precision of the estimates, with significant predictors including age, total fat intake, physical activity, and SES.

**Table 2. Multilevel Bayesian logistic regression analysis of MetS and leading NCD risk factors among 118 public primary school teachers in mid-2019 in Gqeberha, South Africa. Bold values indicate statistically significant results.**

| | Odds ratio (OR) | 95% OR credible interval (CrI): | |
|---|---|---|---|
| | | Lower | Upper |
| **Demographic predictors** | | | |
| **Age (years)** | **1.15** | **1.07** | **1.24** |
| Sex (man vs woman) | 0.33 | 0.06 | 1.57 |
| Race (Coloured vs Black) | 2.11 | 0.56 | 7.73 |
| Race (White vs Black) | <0.01 | <0.01 | 3.67 |
| Education attainment (University degree vs diploma) | 1.23 | 0.39 | 4.13 |
| Education attainment (Postgraduate degree vs diploma) | 0.36 | 0.08 | 1.53 |
| SES[1] (middle[2] vs lower[3]) | 0.44 | 0.14 | 1.31 |
| **SES[1] (higher[4] vs lower[3])** | **0.14** | **0.02** | **0.72** |
| **Physical activity predictor** | | | |
| **MVPA[5] (hours/week)** | **0.73** | **0.55** | **0.95** |
| **Substance use predictors** | | | |
| Tobacco use (yes vs no) | 1.10 | 0.27 | 4.74 |
| Alcohol use (yes vs no) | 1.08 | 0.34 | 3.48 |
| **Nutrient intake predictors** | | | |
| **High total fat (yes vs no)** | **3.37** | **1.03** | **11.96** |
| High saturated fat (yes vs no) | 0.98 | 0.30 | 3.23 |
| High total sugar (yes vs no) | 2.16 | 0.75 | 6.55 |
| High sodium (yes vs no) | 1.08 | 0.35 | 3.45 |

**Legend for Table 2:**

[1]SES: Socioeconomic status based on average monthly household income

[2]SES middle: R 20 001 - R 40 000 per month

[3]SES lower: ≤ R 20 000 per month

[4]SES higher: > R 40 000) per month

[5]MVPA: Moderate-to-vigorous intensity physical activity

## 4. Discussion

This study investigated MetS and leading modifiable risk factors for NCDs among teachers working in under-resourced public primary schools in Gqeberha, South Africa. We found that almost 3 out of 5 teachers met the latest diagnostic criteria for MetS, and almost half failed to meet the WHO physical activity guidelines. Additionally, significant positive associations were observed between MetS and advancing age, consuming a diet high in total fat, engaging in insufficient physical

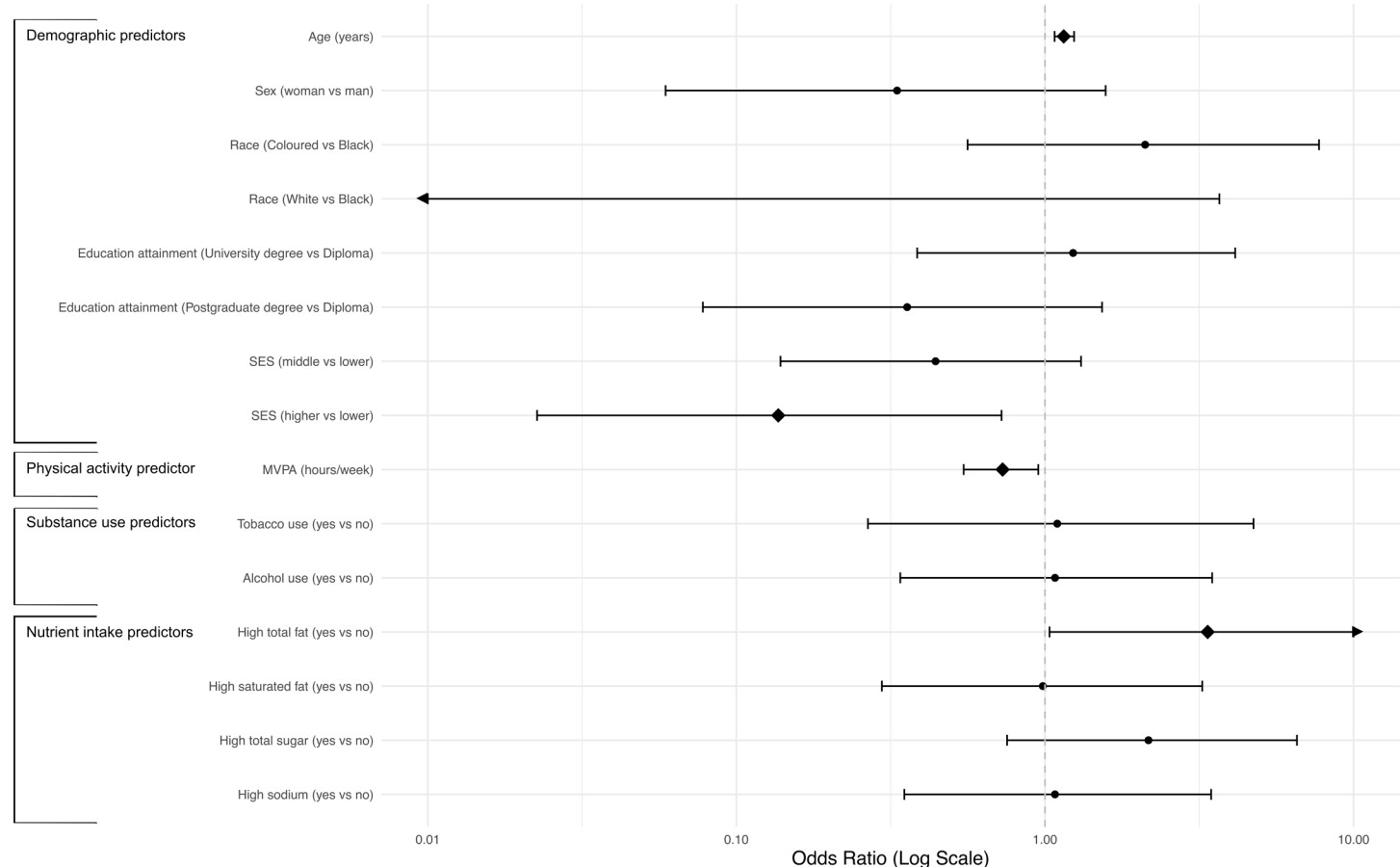

**Fig 2. Multilevel Bayesian logistic regression: association between MetS and NCD risk factors among 118 public primary school teachers in mid-2019 in Gqeberha, South Africa. Legend for Fig 2:** SES: Socioeconomic status based on average monthly household income, SES lower: ≤R 20 000 per month, SES middle: R 20 001 - R 40 000 per month, SES higher: >R 40 000 per month, MVPA: Moderate-to-vigorous intensity physical activity. ◆ Statistically significant predictor,95% credible intervals (CrI) are presented for odds ratios (ORs), Arrows indicate that the 95% CrI for race (White vs Black) and high total fat (yes vs no) extend beyond the x-axis range.

activity, and lower SES. Given the observational nature of the study, the results should be interpreted with caution. The estimated effect sizes indicate the strength of the relationship between MetS and its predictors but do not necessarily imply causal effects. Our findings however underscore a critical health concern within this population with implications for both educators and the public educational environment in South Africa.

Using the latest JIS definition of MetS, 58% (n = 92) of teachers met the diagnostic criteria for MetS [9]. When compared to a recent global meta-analysis, the prevalence found in this study was substantially higher than the global MetS average of 12.5-31.4% [11]. Although national estimates of MetS in South Africa are not available, recent studies conducted with healthcare workers, people diagnosed with human immunodeficiency virus (HIV), and pregnant and postpartum women, found lower MetS prevalence of 18%, 20%, and 29%, respectively [43–45]. Since MetS significantly increases the risk of developing cardiovascular diseases, strokes, and type 2 diabetes, and given that diabetes, hypertension, ischaemic heart disease, and other forms of heart disease were among the top 10 leading natural causes of death in South Africa in 2018, addressing the prevalence and risk factors for MetS is critical to reducing the burden of NCDs and improving public health outcomes in South Africa [46].

Central obesity was the predominant risk factor in our study (78%, n = 107), followed by hypertension (59%, n = 92) and elevated triglycerides (51%, n = 82). Globally, central obesity has also been reported as the most prevalent individual MetS risk factor (45.1%), with even higher prevalence recorded among South African teachers (92%), highlighting central obesity as a key risk, especially in teacher populations [11,47]. Interestingly, previous studies in teacher samples in South Africa highlighted that teachers did not realise that excess body weight increased their risk for NCDs, and hence, did not see the benefits of weight loss [48]. The lack of NCD awareness aligns with our findings, where the majority of the study sample were unaware of their NCD status, with only about 10% of teachers using medication for dyslipidaemia, hypertension, and/or hyperglycaemia, despite many more meeting the criteria and likely requiring pharmacological intervention based on their health risks. Barriers to seeking NCD care in South Africa have, however, been documented [49]. Barriers affecting women, in particular, include low self-awareness of NCD status, compounded by supply-side barriers such as inadequate knowledge about available healthcare services, resource constraints, and inefficiencies in the healthcare system. These challenges highlight the critical need for improved health education and healthcare awareness, particularly as NCDs and physical inactivity increase with age, with women – the majority of primary school teachers in South Africa – being at higher risk for both obesity and physical inactivity.

Globally, a key driver for promoting physical activity is reducing the burden of NCDs. To achieve this, the WHO physical activity guidelines recommends 150–300 min of moderate, or 75–150 min of vigorous intensity physical activity, or an equivalent combination, per week, for all adults [35]. The 2022 global physical activity estimates indicate, however, that a third of adults (31.3%) do not meet this recommendation [50]. Within our study sample, 44% of teachers did not reach a minimum of 150 min of MVPA per week, and achieving this level of physical activity was found to be protective against the development of MetS. The latest age-standardised national physical activity levels in South Africa align with our findings, where 41% of men and 48% of women failed to meet the physical activity guidelines [50]. This level of inactivity is not only worrisome in itself, but one of the most consistent findings in physical activity research indicates age-related decline [51]. Since the majority of teachers participating in this study were aged between 45 and 64 years, a further sharp decline in their physical activity is projected as they reach 65 years, contributing to further NCD risk.

Tobacco use of 15% in our study sample is below the national average of 20.2% [52]. Although the South African population consume more alcohol than the global and African averages, we found that one third (35%) of teachers in our sample reported any alcohol use [53]. The lower substance use in our study can be attributed to several socio-professional factors, including cultural and religious values, teachers perceiving themselves as role models within their schools and communities, as well as heightened awareness of the negative consequences of substance abuse, influenced by the social challenges prevalent in their school's surrounding communities. While substance use is known to increase the risk of MetS and NCDs, no clear association with MetS was observed in this study. This finding should not diminish the importance of addressing substance abuse as a broader public health concern, particularly in lower socioeconomic communities.

Key dietary risks identified by the WHO, including high intakes of total fat, saturated fat, sugar, and sodium, were prevalent in our study population. More than half of the teachers (53%) consumed diets high in total fat, 42% in saturated fat, 64% in sugar, and 30% in sodium [39]. These components directly contribute to NCD risk, especially overweight, dyslipidaemia, hypertension, and hyperglycaemia. Low nutrition literacy has been described within South African educators teaching in under-resourced schools, where a relatively small proportion of educators knew what a balanced diet consisted of and only 15% knew that one should eat five or more portions of fruit and/or vegetables per day [48]. These findings, coupled with the fact that high total fat was a significant predictor of MetS in our study sample and central obesity was the most prevalent individual MetS component, underscore the need for urgent and targeted interventions to improve dietary patterns among educators. This can be achieved through nutrition education programmes that address knowledge gaps and raise awareness of the risks associated with excessive intake of fat, sugar, and sodium, while linking dietary habits to maintaining a healthy body weight. Although initiatives such as the sugar tax, mandatory sodium reduction, and

nutritional labelling have been introduced in South Africa, challenges such as limited access to affordable healthy foods and a reliance on processed, energy-dense options highlight the ongoing need for education and structural interventions to support healthier eating behaviours, which are critical for reducing the burden of NCDs.

Our study further identified higher monthly household income as a significant protective factor against the development of MetS. This finding underscores the critical role of social determinants of health in influencing health outcomes. Consistent with recent research, which demonstrated that economic stability and other social determinants of health fully explained disparities in cardiovascular disease mortality among different population groups in America, our results suggest that financial resources significantly mitigate MetS risk [54]. While individual lifestyle factors such as physical activity, diet, and substance use are important for reducing NCD risk, marginalized populations may require additional interventions that address broader social determinants. Supporting this notion, a study analysing 20 years of American health data found that improvements in diet and physical activity behaviours significantly reduced chronic diseases among non-Hispanic White individuals, but had a less pronounced effect among non-Hispanic Black and Hispanic populations [55]. This suggests that broader social and environmental conditions may limit the effectiveness of traditional lifestyle interventions in certain population groups. Hence, to effectively mitigate MetS and other NCDs in under-resourced settings, it is essential to implement comprehensive strategies that not only promote healthy behaviours but also address the underlying social and economic factors that contribute to health inequities.

Future research should explore whether the high prevalence of MetS observed in this study reflects a broader national trend among teachers in South Africa. This could be achieved through cost-effective approaches such as secondary data analysis or targeted subnational studies. Further incorporating covariates such as mental health, which impacts metabolic health, physical activity levels, and overall wellbeing could provide a more comprehensive understanding of the determinants of MetS and inform targeted interventions. Additionally, given the substantial burden of MetS in this population, research evaluating potential interventions would be essential to inform effective prevention strategies. Effective interventions should extend beyond individual-level lifestyle modifications to incorporate community-based, environmental, occupational, policy, and health system approaches. Such interventions should promote regular NCD screening and education strategies to raise awareness, facilitate early detection and treatment, and support individuals and communities to adopt and sustain healthier behaviours, especially around physical activity and diet. Prioritising teachers' health is not only essential for their wellbeing but also for improving educational outcomes and fostering healthy habits among the children they teach. This is especially critical in low-resourced settings, where teachers often serve as the primary stable role models. Failing to address the burden of NCDs and MetS among teachers risks a cascading crisis with severe implications for both the education sector and future generations.

### 4.1. Strengths and limitations

The findings of our study should be interpreted in the light of the following limitations. First, HbA1c was used as a surrogate measure of glucose levels instead of plasma fasting blood glucose measurements due to resource constraints. Fasting plasma glucose values and HbA1c have, however, shown comparable performance in the South African context [9,24]. Second, participants wore accelerometers for a minimum of 8 h per day, limiting the ability to capture their complete daily physical activity. The 8-h minimum wear-time threshold is, however, a commonly accepted criterion and a minimum of 8 h of waking time physical activity captures a substantial portion of daily physical activity. Third, the 24-h dietary recall was conducted only once, making it susceptible to recall bias and social desirability bias. Fourth, mental health and disability were not included as covariates, which may have influenced the observed associations as both covariates can impact metabolic health and physical activity levels. Fifth, this study is observational and cross-sectional in nature, requiring caution in interpreting the results. While the associations observed provide valuable insights, causal effects cannot be directly inferred without accounting for confounding factors and theoretical assumptions. Finally, while the study included 8 schools representative of under-resourced public primary schools in South Africa, the relatively small and context-specific

sample restricts the generalisability of the findings to the broader population of public school teachers nationwide. Although statistical methods were employed to address sample limitations, the results remain closely tied to the characteristics of the observed sample.

Despite these limitations, the study has several strengths that enhance its contribution to the field. It provides a comprehensive cardiometabolic health assessment of primary school teachers, an occupational group that is vital, yet often overlooked, in the South African context. Moreover, the use of device-based physical activity measurement is not commonly used in LMICs, especially in adult populations. Accelerometer measured physical activity provides precise and empirical insights into activity levels, effectively overcoming the limitations commonly associated with self-reported physical activity data.

## 5. Conclusion

Our study highlights a substantial burden of NCD risk among public primary school teachers in under-resourced schools in Gqeberha, South Africa. We found a high prevalence and severity of MetS, primarily driven by obesity, hypertension, and elevated triglycerides. Physical inactivity, a diet high in total fat, and lower SES were significant contributors to MetS risk. These findings underscore the urgent need for targeted public health interventions that not only promote healthier lifestyle behaviours but also address structural barriers to health equity. Limited healthcare awareness, nutritious food, and safe opportunities for physical activity in low-resourced communities exacerbates the risk of NCDs, highlighting the necessity for comprehensive, sustainable, and multi-level interventions.

## Supporting information

**S1 Table. List of abbreviations.**
(DOCX)

**S1 Fig. Distribution of MetS components and combinations by severity count among 149 public primary school teachers in mid-2019 in Gqeberha, South Africa. Legend for S1 Fig**: MetS: Metabolic syndrome, WC: Central obesity (waist circumference with imputed values), HTN: Hypertension, TG: High triglyceride, HDL: Low high density lipoprotein cholesterol, HGL: Hyperglycaemia.
(TIFF)

## Acknowledgments

The authors are grateful to the teachers for their participation in the study and the education authorities for their support. We would like to thank Prof. Dr. Ilse Truter (School of Clinical Care and Medicinal Sciences, Nelson Mandela University) for providing insights into pharmacology and diagnosis. Thanks are also addressed to Accelting for assisting with the open-source accelerometry analysis methodology.

## Author contributions

**Conceptualization:** Nandi Joubert, Jürg Utzinger, Markus Gerber, Cheryl Walter.

**Data curation:** Nandi Joubert, Larissa Adams.

**Formal analysis:** Nandi Joubert, Jan Hattendorft.

**Funding acquisition:** Nandi Joubert, Annelie Gresse, Ivan Müller, Nicole Probst-Hensch, Harald Seelig, Peter Steinmann, Rosa du Randt, Uwe Pühse, Jürg Utzinger, Markus Gerber, Cheryl Walter.

**Investigation:** Nandi Joubert, Larissa Adams, Jan Degen, Danielle Dolley, Siphesihle Nqweniso, Rosa du Randt, Cheryl Walter.

**Methodology:** Annelie Gresse, Ivan Müller, Nicole Probst-Hensch, Harald Seelig, Peter Steinmann, Rosa du Randt, Uwe Pühse, Jürg Utzinger, Markus Gerber, Cheryl Walter.

**Project administration:** Nandi Joubert, Larissa Adams, Rosa du Randt, Cheryl Walter.

**Supervision:** Rosa du Randt, Jürg Utzinger, Markus Gerber, Cheryl Walter.

**Visualization:** Nandi Joubert, Jan Hattendorft.

**Writing – original draft:** Nandi Joubert.

**Writing – review & editing:** Nandi Joubert, Larissa Adams, Jan Hattendorft, Jan Degen, Danielle Dolley, Annelie Gresse, Ivan Müller, Siphesihle Nqweniso, Nicole Probst-Hensch, Harald Seelig, Peter Steinmann, Rosa du Randt, Uwe Pühse, Jürg Utzinger, Markus Gerber, Cheryl Walter.

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
