## [Decision Letter · Decision Letter 0]

9 Mar 2025

PGPH-D-25-00167

Metabolic syndrome, physical activity, and modifiable risk factors of non-communicable diseases among teachers in under-resourced schools in South Africa: Baseline findings from the KaziHealth workplace health intervention

Dear Dr. Joubert,

Thank you for submitting your manuscript to PLOS Global Public Health. After careful consideration, we feel that it has merit but does not fully meet PLOS Global Public Health’s publication criteria as it currently stands. Therefore, we invite you to submit a revised version of the manuscript that addresses the points raised during the review process.

Please concisely present conclusion based on the study findings in brief. 

We look forward to receiving your revised manuscript.

Kind regards,

Dr Buna Bhandari

Academic Editor

Journal Requirements:

  1. We have amended your Competing Interest statement to comply with journal style. We kindly ask that you double check the statement and let us know if anything is incorrect.  2. We notice that your supplementary figure S1 is uploaded with the file type 'Figure'. Please amend the file type to 'Supporting Information'. Please ensure that each Supporting Information file has a legend listed in the manuscript after the references list.  3. In the online submission form, you indicated that “The data underlying this study will be made publicly available in Zenodo upon acceptance of the manuscript for publication to ensure alignment with the final published version. A DOI for the dataset will be provided in the final published article. For peer review purposes, the data are available from the corresponding author upon request.”.  All PLOS journals now require all data underlying the findings described in their manuscript to be freely available to other researchers, either 1. In a public repository, 2. Within the manuscript itself, or 3. Uploaded as supplementary information. This policy applies to all data except where public deposition would breach compliance with the protocol approved by your research ethics board. If your data cannot be made publicly available for ethical or legal reasons (e.g., public availability would compromise patient privacy), please explain your reasons by return email and your exemption request will be escalated to the editor for approval. Your exemption request will be handled independently and will not hold up the peer review process, but will need to be resolved should your manuscript be accepted for publication. One of the Editorial team will then be in touch if there are any issues.

Additional Editor Comments (if provided):

Please concisely present conclusion based on the study findings in brief. 

Reviewers' comments:

Reviewer's Responses to Questions

**Comments to the Author**

1. Does this manuscript meet PLOS Global Public Health’s publication criteria ? Is the manuscript technically sound, and do the data support the conclusions? The manuscript must describe methodologically and ethically rigorous research with conclusions that are appropriately drawn based on the data presented.

Reviewer #1: Partly

Reviewer #2: Yes

2. Has the statistical analysis been performed appropriately and rigorously?

Reviewer #1: No

Reviewer #2: I don't know

3. Have the authors made all data underlying the findings in their manuscript fully available (please refer to the Data Availability Statement at the start of the manuscript PDF file)?

Reviewer #1: Yes

Reviewer #2: Yes

4. Is the manuscript presented in an intelligible fashion and written in standard English?

Reviewer #1: Yes

Reviewer #2: Yes

5. Review Comments to the Author

Reviewer #1: The study subject and context is an important on-going area of research in chronic illnesses and their modifiable risk factors. This study represents the baseline to establish a reference point prior to implementing an educational/behavioural intervention programme. This welcomed with enthusiasm cosidering that findings thereafter would benefit a vast at risk population. However, as much as this phase of the study represents the reference point to evaluate the outcome of the intervention, is very important that it is conceived in the highest standard and rigor of scientific principles for validity of the proof of concept underpinning the dynamics of the problem phenomenon for which the intervention has been designed. There are few observations made in this review to contribute significantly to strenthening the study argument and they are as follows:

1. The title “Metabolic syndrome, physical activity, and modifiable risk factors for non-communicable diseases among teachers in under-resourced schools in South Africa: Baseline findings from the KaziHealth workplace health intervention” would benefit greatly through modification because the present title has not placed the variables in the correct order of explanatory variables and outcome variable. Explanatory variables should appear first because they explain the dynamics of the outcome variable. Where "Metabolic syndrome" is placed first appear to be an explanatory variable whereas it is a clinical outcome. Again, physical activity is a modifiable risk factor for non-communicable diseases and therefore should be remove. Suggested correction should be: “Predictors of metabolic syndrome among teachers in under-resourced schools in South Africa: Baseline findings from the KaziHealth workplace health intervention”. With this title rendered this way, the physical activitiy, dietary concerns and substance use becomes embaded as the modifiable risk factors and their antecedent factors collectively covered by the word "predictors".

Justification for this revision is that Metabolic syndrome is the term generally used for describing cluster of non-communicable diseases that explain biochemical disfunction of metabolic pathways leading poor glucose control that trigger blood pressure control among others. The major modifiable risk factor for these diseases is usually a risk behaviour such as physical activity, tobacco-use and diet, therefore risk factors for non-communicable diseases would obviously be primarily behavioural and their antecedents. There is observed syntax issue with the original title. The “Baseline findings from the KaziHealth workplace health intervention” is actually redundant and makes the reading of the title unnecessarily long.

2. Unfortunately the study has not been guided by sound public health principles to facilitate elucidating the dynamics of the problem phenomenon. The entire study is not guided by behaviour theoretical framework and therefore weak and unable to provide elucidation of the dynamics of the identified problem phenomenon. This study without any theoretical framework cannot hypothesize the dynamics of the study. Contemporary high grade research are not conceived this way. Kindly consider the recommended revision.

Reviewer #2: Thanks you for a well written paper on an important topic.

My comments are intended to strengthen the article, so please accept them in this spirit:

1. Covariates not considered: health insurance, mental health, disability - should be mentioned in 'limitation'

2. Please explain how the reader should interpret the Odds Ratio, given your statement (line 619) re: cross-sectional study design

3. The conclusion would beneffit from recommendations for further research in the field.

General comment: there were more European than African authors. The lead autheor is based at a European institution, while only one senior author is based at an African institution. This is against the spirit of global health from a decolonial perspective. How can this be addressed?

6. PLOS authors have the option to publish the peer review history of their article (what does this mean? ). If published, this will include your full peer review and any attached files.

**Do you want your identity to be public for this peer review?** For information about this choice, including consent withdrawal, please see our Privacy Policy .

Reviewer #1: **Yes: ** Nnodimele Onuigbo ATULOMAH

Reviewer #2: **Yes: ** Tasleem Ras

---

## [Decision Letter · Decision Letter 1]

6 May 2025

Predictors of metabolic syndrome among teachers in under-resourced schools in South Africa: Baseline findings from the KaziHealth workplace health intervention

PGPH-D-25-00167R1

Dear Joubert,

We are pleased to inform you that your manuscript 'Predictors of metabolic syndrome among teachers in under-resourced schools in South Africa: Baseline findings from the KaziHealth workplace health intervention' has been provisionally accepted for publication in PLOS Global Public Health.

Best regards,

Buna Bhandari

Academic Editor

Reviewer Comments (if any, and for reference):

Reviewer's Responses to Questions

**Comments to the Author**

1. If the authors have adequately addressed your comments raised in a previous round of review and you feel that this manuscript is now acceptable for publication, you may indicate that here to bypass the “Comments to the Author” section, enter your conflict of interest statement in the “Confidential to Editor” section, and submit your "Accept" recommendation.

Reviewer #1: All comments have been addressed

Reviewer #2: All comments have been addressed

2. Does this manuscript meet PLOS Global Public Health’s publication criteria ? Is the manuscript technically sound, and do the data support the conclusions? The manuscript must describe methodologically and ethically rigorous research with conclusions that are appropriately drawn based on the data presented.

Reviewer #1: Yes

Reviewer #2: Yes

3. Has the statistical analysis been performed appropriately and rigorously?

Reviewer #1: Yes

Reviewer #2: I don't know

4. Have the authors made all data underlying the findings in their manuscript fully available (please refer to the Data Availability Statement at the start of the manuscript PDF file)?

Reviewer #1: Yes

Reviewer #2: Yes

5. Is the manuscript presented in an intelligible fashion and written in standard English?

Reviewer #1: Yes

Reviewer #2: Yes

6. Review Comments to the Author

Reviewer #1: (No Response)

Reviewer #2: The recommendations for future research are still not there, though I acknowledge the recommendations for developing interventions.

7. PLOS authors have the option to publish the peer review history of their article (what does this mean? ). If published, this will include your full peer review and any attached files.

**Do you want your identity to be public for this peer review?** For information about this choice, including consent withdrawal, please see our Privacy Policy .

Reviewer #1: **Yes: ** Nnodimele Onuigbo ATULOMAH

Reviewer #2: **Yes: ** Tasleem Ras
